## TOPICAL REVIEW

# Ultrastructure of astrocytes using volume electron microscopy: A scoping review

Vanessa Chiappini[1,2,3], Maria Fernanda Veloz Castillo[1,2,4], Francesco Biancardi[3], Ferdinando Di Cunto[1,2], Pierre J. Magistretti[4], Alessandro Vercelli[1,2], Marco Agus[5] and Corrado Calì[1,2] 🆔

[1]*Department of Neuroscience, University of Turin, Turin, Italy*
[2]*Neuroscience Institute Cavalieri Ottolenghi, Orbassano, Italy*
[3]*Carl Zeiss Microscopy, Oberkochen, Germany*
[4]*Biological and Environmental Sciences and Engineering Division, King Abdullah University of Science and Technology, Thuwal, Saudi Arabia*
[5]*College of Science and Engineering, Hamad bin Khalifa University, Doha, Qatar*

Handling Editors: Nathan Schoppa & Valentina Mosienko

The peer review history is available in the Supporting Information section of this article (https://doi.org/10.1113/JP287455#support-information-section).

**Abstract figure legend** A complete overview on astrocyte ultrastructure with volume electron microscopy. The morphology of astrocytes under physiological and pathological conditions plays a role in brain homeostasis, synaptic activity and structural support, but their morphological complexity and heterogeneity are still poorly quantified. Electron microscopy (EM) is a valuable source for performing structural analysis of these complex cells, given the size of their smaller processes, up to two orders of magnitude below visible light resolution. Moreover, with the rise of volume EM (vEM), the analysis of extensive 3-D samples has become more accessible. This review summarizes all the papers that study astrocyte ultrastructure with vEM, focusing on the CNS regions and the animal models studied, as well as the vEM techniques employed, highlighting their advantages, limitations and applicability. The type of segmentation, whether manual or automated, is detailed, along with references to commonly used software. Finally, the focus is placed on the specific astrocyte-related biological questions addressed through vEM. ssTEM, serial section transmission electron microscopy; ssET, serial section electron tomography; FIB-SEM, focused ion beam-scanning electron microscopy; SBF-SEM, serial block face-scanning electron microscopy; ATUM-SEM, automated tape-collecting ultra microtome-scanning electron microscopy. Created with BioRender.com.

**Abstract** The morphological features of astrocytes are crucial for brain homeostasis, synaptic activity and structural support, yet remain poorly quantified. As a result of the nanometre-sized cross-section of neuropil astrocytic processes, electron microscopy (EM) is the only technique availabe to date capable of revealing their finest morphologies. Volume EM (vEM) techniques, such as serial block-face or focused ion beam scanning EM, enable high-resolution imaging of large fields and allow more extensive 3-D model analyses, revealing new astrocytic morphological features. This scoping review aims to summarize the state of the art of astrocyte ultrastructural analysis. This review included 45 of 439 non-duplicated articles from a Pubmed search, categorizing studies by research focus, animal models, brain region, vEM techniques and segmentation methods. By answering classical questions such as volume, surface area, branching complexity and synaptic ensheathment reported in the literature, this work is a valuable resource for scientists working on structural biology or computational neuroscience.

(Received 29 November 2024; accepted after revision 12 February 2025; first published online 5 March 2025)

**Corresponding author** M. Agus: College of Science and Engineering, Hamad bin Khalifa University, LAS Building, Education City, Doha, Qatar. Email: magus@hbku.edu.qa; C. Calì: Department of Neuroscience, University of Turin, Corso Massimo D'Azeglio 52, Turin 10126, Italy. Email: corrado.cali@unito.it

## Introduction

Astrocytes are the most abundant glial cell population in the CNS. However, their percentage is still debated, estimated between 20% and 40% of the total number of CNS cells, with significant variability between species and brain regions (Herculano-Houzel, 2014). Among their many crucial tasks, they are fundamental players in maintaining brain homeostasis by regulating cerebral blood flow, preserving the integrity of the blood–brain barrier and facilitating the clearance of metabolic waste from brain tissue via the glymphatic system (Magistretti & Allaman, 2022). Astrocytes actively influence synaptic activity and plasticity by being involved in processes such as synapse formation, maintenance, maturation and synaptic transmission (Akther & Hirase, 2022; Santello et al., 2012). They are essential for neuronal metabolic support because they reuptake and convert extracellular glutamate into recyclable glutamine at the same time as providing energy to neurons by metabolizing glycogen into lactate (Attwell et al., 2010; Pellerin & Magistretti, 1994; Voutsinos-Porche et al., 2003).

Astrocytes are classically categorized, based on non-human mammals studies, as protoplasmic in grey matter and fibrous in white matter, with each group having specific roles. Protoplasmic astrocytes have a highly branched, bushy morphology with complex spongiform shapes. They exert multiple roles in synaptic development and function (Emsley & Macklis, 2006; Miller & Raff, 1984). Fibrous astrocytes are often more elongated, have fewer and longer processes, and have a more distinct role in support of long-distance transmitting fibres (Köhler et al., 2019; Miller & Raff, 1984). Nonetheless, the analysis of primate and human tissue further highlighted the heterogeneity of these cells based on their molecular diversity and possibly reflecting their morphological diversity (Viana et al., 2023). For example, astrocytes with unique features were identified in human and non-human primates (NHPs), such as long varicose projecting astrocytes from cortical layer V or subpial interlaminar astrocytes (Oberheim et al., 2009). Specialized astroglial cells such as Müller cells in the retina and Bergmann glial cells in the cerebellum add further complexity to astrocytic functions.

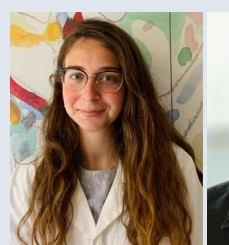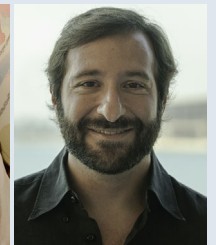

**Vanessa Chiappini** is a PhD student in Complex Systems for Quantitative Biomedicine at the University of Turin. She was trained as a Biomedical Engineer at Politecnico di Torino. After a scholarship at (Orbassano, Italy), she became interested in Neuroscience, and she is now becoming expert in quantitative volume electron microscopy. **Corrado Calì**, is an associate professor of Human Anatomy at the University of Torino. He was trained as a biomedical engineer at Politecnico di Torino. He completed his MSc in 2006 at EPFL, where he developed an interest in neuroscience. He earned his PhD in 2012 at UNIL, focusing on astroglial cells and synaptic transmission. After postdoctoral work in 3-D electron microscopy techniques, he joined KAUST to study astrocyte-neuron metabolic support. His team pioneered virtual reality techniques in neuroscience. In 2020, he became an assistant professor at the University of Torino and founded Intravides, a startup developing augmented reality tools for neurosurgery.

Astrocytes have a star-like shape, with a compact cell body and a variable number of main processes stemming from their soma. Although there is no clear ontology for smaller processes, they can be classified by their size using a terminology deriving from literature: branches, branchlets, leaflets and endfeet (Baldwin et al., 2023; Semyanov & Verkhratsky, 2021). The stem processes are referred to as branches, whereas the secondary and tertiary processes are named branchlets. Leaflets are also known as peripheral fine processes or peri-synaptic astrocytic processes (PAPs), and they represent the finest processes. Endfeet, on the other hand, are specialized and polarized astrocytic processes that come into contact with blood vessels and pericytes and are the only morphologically recognizable trait of astrocytes (Calì et al., 2019; Khakh & Sofroniew, 2015).

Diffraction-limited light microscopy has been widely employed to study astrocyte morphology; however, understanding how branchlets and PAPs extend from branches and measure their parameters is not possible with these imaging techniques because they do not have the necessary spatial resolution (Octeau et al., 2018). Moreover, glial fibrillary acidic protein (GFAP) immunostaining, the most common marker used to identify astrocytes, highlights only ~15% of the total astrocytic volume (Bushong et al., 2002). Hence, the only technique able to visualize diffraction-limited processes, and consequently their full morphology, is electron microscopy (EM). A shorter wavelength of electrons compared to photons enables EM to achieve nanometric resolution. Knoll & Ruska (1932) developed the first transmission electron microscope (TEM) in 1932, immediately emerging as a powerful biological imaging tool. EM has undergone revolutionary advancements, particularly in the past two decades with the rise of volume EM (vEM) techniques (Knott & Genoud, 2013).

These cutting-edge methods facilitate the imaging of tissue volumes for 3-D ultrastructural analysis by automating the acquisition of thousands of serial electron micrographs, thereby streamlining the process, enhancing reliability and overcoming the two rate-limiting steps of manual sectioning and imaging. These recent advancements in vEM paved the way for breakthroughs in the structural and ultrastructural analysis of glial cells, and astrocytes in particular.

vEM on astrocytes can shed light on questions related to their spatial distribution, synaptic coverage and regional heterogeneity, to name but a few (Benedusi et al., 2024). The growing field of computational neuroscience is becoming increasingly hungry for such structural and ultrastructural information, which is of the utmost importance to feed simulation algorithms. Furthermore, a growing awareness that the compartmentalization of astrocytes has fundamental functional implications strongly encourages a shift from point-based models to geometrical ones (Denizot et al., 2022). Many of the data necessary to populate such models can be found in the existing literature, following an approach already addressed by the Human Brain Project (https://www.humanbrainproject.eu) consortium. Such information can be fetched from review articles, but review papers on vEM (Peddie & Collinson 2014; Peddie et al., 2022; Titze & Genoud, 2016) provide valuable insights into the technical developments in the field, without focusing on a specific biological topic. Literature on vEM on astrocytes was revised in a review by Calì (2017) but lacks updated information. Here, we fill the existing gap in the systematic analysis of literature and report the major ultrastructural findings on astrocytes, analyse the different animal models studied, and compare the vEM and image segmentation techniques employed. This scoping review could be a valuable resource to retrieve relevant literature on the state of the art on astrocyte reconstruction and quantitative ultrastructural analysis.

## Methods

The Preferred Reporting Items for Systematic Reviews and Meta-Analyses extension for Scoping Reviews (PRISMA-ScR) (Fig. 1) (Tricco et al., 2018) was used for this scoping review. The advanced research was conducted between November and December 2023 on Pubmed, using the keywords: ((astrocyte) AND (serial section EM)) OR ((astrocyte) AND (three-dimensional EM)) OR ((astrocyte) AND (volume EM)) OR ((astrocyte) AND (3D reconstruction)) OR ((glia) AND (3D EM)). Exclusion criteria included publications in the format of poster abstracts, editorial letters or conference papers. Studies with irrelevant titles and abstracts were detected and removed. Furthermore, we excluded studies on single-section electron micrographs or those that did not focus specifically on astrocytes. Studies cited in peer-reviewed review papers were also screened for broader inclusion.

## Results

In total, 439 distinct titles were reviewed and 45 papers were selected for the scoping review. All the studies examined in this review are presented in Tables 1 and 2, categorized according to their use of either analog or digital acquisition mode.

**vEM techniques used to study astrocyte ultrastructure.**
Subsequent to the development of serial section EM, many researchers have strived to study the 3-D ultrastructure of astrocytes to infer a better understanding of their functions based on their morphology. These studies

can be classified based on the EM technique used to image the samples. Manual sectioning of slices combined with TEM, named serial section TEM (ssTEM), was the first vEM technique ever developed. Subsequent to the pioneering work on astrocyte morphology by Wolff (1965), ssTEM has been the most widely used technique for imaging astrocytes with vEM (Fig. 2). A similar approach to ssTEM is the more recent serial section SEM (ssSEM), which was used by Watanabe et al. (2010) to study the organization of the perivascular glial limiting membrane. ssTEM and ssSEM involve manual sectioning and collection of ultrathin sections, which is a process that is time-consuming, laborious and error-prone. Compared to ssTEM, the ssSEM approach used by Watanabe et al. (2010) has the advantage of using thicker sections, which are more resistant and easier to handle, although the sample processing might be more labour intensive, requiring, for example, an ion etcher to remove the embedding resin. Because sectioning is a rate-limiting step for vEM, other approaches combining EM imaging for resolution, without the need for sectioning, have been developed. For example, Hama et al. (2004) used electron tomography, where a single, thicker section is reconstructed from many tilted series of TEM images.

However, the longer time and energy exposure needed might lead to tissue distortion or bleaching. In the early 2000s, novel techniques were developed allowing the automation of sectioning and imaging processes. The focused ion beam scanning electron microscopy (FIB-SEM) technique allows imaging of small volumes (to ~10,000 $\mu m^3$) by automatically milling thin layers (in the nanometre range) of material from the block-face gaining isotropic resolution (Titze & Genoud, 2016), which makes it very suitable for studying the neuropil. Another automated technique is the serial block face scanning electron microscopy (SBF-SEM). In these setups, the SEM chamber size allows to fit an automated ultramicrotome, which cuts the block face. The acquired field of view (FOV) is not limited, allowing acquisitions of hundreds of cubic micrometres (Calì et al., 2019). Compared to FIB-SEM, the physical sectioning limits section thickness to 20 nm at best. Nevertheless, this instrument is an excellent compromise between acquired FOV, voxel resolution and ease of use, making it a very versatile system. The work by Sweeney et al. (2017) used axial scanning transmission electron microscopy tomography (Axial-STEM tomography) for imaging a 1 $\mu$m thick sample. This unique technique, notably

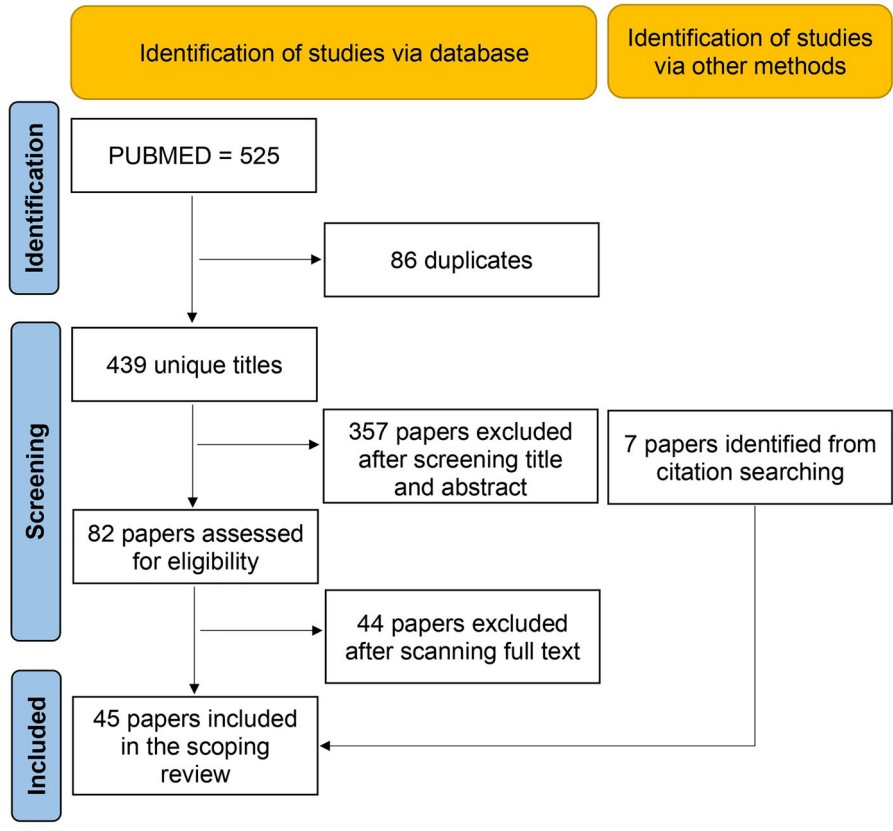

**Figure 1. PRISMA-ScR**
Preferred Reporting Items for Systematic Reviews and Meta-Analyses extension for Scoping Reviews chart for the identification of papers included in the scoping review.

**Table 1. Key studies on astrocytes ultrastructure with analogical volume electron microscopes and graphical reconstruction.**

| Model/region | EM technique | Section thickness (nm) | Segmentation | Labelled (sub)cellular structures | Paper |
|---|---|---|---|---|---|
| Rabbit and mouse: temporal or parietal cerebral cortex, the subcortical medulla, the midbrain and cerebellum | ssTEM | 50 | Manual | Astrocytic processes | Wolff (1965) |
| Toad: brain and spinal cord | ssTEM | 40–80 | Manual | Astrocyte, oligodendrocyte and microglia | Stensaas (1968) |
| Cat: spinal cord | ssTEM | – | Manual | Motor neuron, astrocyte | Poritsky (1969) |
| Rat: ventrobasal nuclear complex | ssTEM | 50 | Manual | Astrocytic processes | Spacek & Lieberman (1971) |
| Rabbit: parietal cerebral cortex and corpus callosum | ssTEM | 100 | Manual | Astrocytic and oligodendrocytic processes | Spacek (1970) |
| Mouse: cerebellar cortex | ssTEM | 120 | Manual | Tripartite synapses | Spacek (1985) |

distinct from TEM tomography, is not susceptible to chromatic aberration because of the strategic placement of the objective lens ahead of the specimen (Sousa et al., 2010), but requires special equipment, limiting its diffusion. A recent connectomic study, adopting the Automated Tape-collecting Ultra Microtome (ATUM) technique coupled with multibeam SEM, enabled the acquisition of a substantial volume of 1 mm³, entailing the sectioning of over 5000 slices at 30 nm thickness each. This process allowed the unprecedented visualization

of more than 50,000 brain cells, including astrocytes, marking a pivotal milestone in the field (Shapson-Coe et al., 2021). Figure 2 shows the progression of EM imaged volume of glia broken down by year and by vEM technique (only 18 papers reported this information). Although we must state that these data reflect exclusively the articles analysed in this review, we can report an over-all increase in imaged volumes, particularly starting from 2017 (dotted bold line). From 2017 onwards, imaging volumes obtained with ssTEM have increased by over two orders of magnitude, whereas those obtained with FIB-SEM, a technique that inherently has imaging volume limitations, have increased by one order of magnitude. From 2019, SBF-SEM and ATUM-SEM have been as well employed to study astrocytes: as seen previously, these techniques have enabled the acquisition of larger volumes, up to 1 mm³.

**Methods and software tools used for image segmentation and reconstruction of astrocytes.** Scaling up of imaged volumes using EM moved the technical rate-limiting step from sectioning and imaging to data processing. Image segmentation is challenging as a result of sample variability induced by individual imaging conditions, tissue contrast, structural diversity and large data volumes. Manual segmentation by experts is the most reliable method, but is also extremely labour intensive. Automated segmentation methods, although advanced, have yet to reach human-level accuracy, and no fully reliable solution exists. Researchers must choose between manual, semi-automated or fully automated approaches based on segmentation complexity, data

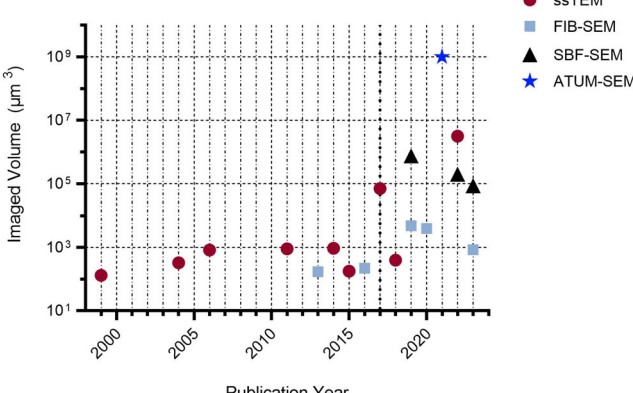

**Figure 2. Maximum volume of imaging per years**
Maximum volume of imaging per years in the analysed papers, highlighting the vEM technique employed. ATUM-SEM, automated tape-collecting ultra Microtome-scanning electron microscopy; FIB-SEM, focused ion beam-scanning electron microscopy; SBF-SEM, serial block face-scanning electron microscopy; ssTEM, serial section transmission electron microscopy; vEM, volume electron microscopy.

**Table 2. Key studies on astrocytes ultrastructure with digital volume electron microscopes.**

| Model/region | EM technique | Resolution | Section thickness (nm) | Segmentation | Labelled (sub)cellular structures | Paper |
|---|---|---|---|---|---|---|
| Rat: stratum radiatum of the hippocampal CA1 region | ssTEM | – | 55 | Manual | Tripartite synapses | Spacek & Harris (1998) |
| Rat: stratum radiatum of the hippocampal CA1 region | ssTEM | – | – | Manual | Tripartite synapses | Ventura & Harris (1999) |
| Mouse: cerebellum | ssTEM | – | 60 | Manual | Appendage of Bergmann glial cell | Grosche et al. (1999) |
| Rat: stratum radiatum of the hippocampal CA1 region | ssTEM | – | 50 | Manual | Tripartite synapses | Spacek & Harris (2004) |
| Rat: dentate gyrus | Electron tomography | – | 2° degrees between tiles | Manual | Astrocytic processes | Hama et al. (2004) |
| Mouse: C2 barrel cortex | ssTEM | – | 60 | Manual | Tripartite synapses | Genoud et al. (2006) |
| Rat: stratum radiatum of the hippocampal CA1 region | ssTEM | – | – | Manual | Tripartite synapses | Witcher et al. (2007) |
| Rat and mouse: cerebral cortex | ssSEM | – | 100–200 | Manual | Perivascular glial limiting membrane | Watanabe et al. (2010) |
| Rat: stratum lacunosum moleculare of the hippocampal CA1 region | ssTEM | – | 45 | Manual | Perivascular astrocyte and endfeet, pericytes | Mathiisen et al. (2010) |
| Human: stratum radiatum of the hippocampal CA1 region | ssTEM | – | 45 | Manual | Tripartite synapses | Witcher et al. (2010) |
| Rat: nucleus Tractus solitarii of the caudal medulla | ssTEM | – | 80 | Manual | Tripartite synapses | Chounlamountry & Kessler (2011) |
| Mouse: optic nerve | FIB-SEM | 7.5 | 30 | Manual | Astrocytic processes, oligodendrocyte and its mitochondrial network | Schertel et al. (2013) |
| Rat: lateral amygdala | ssTEM | – | 45 | Manual | Tripartite synapses | Ostroff et al. (2014) |
| Rat: somatosensory cortex (layer IV) | ssTEM | – | 50–60 | Manual | Dendrites, tripartite synapses | Rollenhagen et al. (2015) |

*(Continued)*

**Table 2. (Continued)**

| Model/region | EM technique | Resolution | Section thickness (nm) | Segmentation | Labelled (sub)cellular structures | Paper |
|---|---|---|---|---|---|---|
| Rat: stratum radiatum of the hippocampal CA1 region | ssTEM | 2 | 50–60 | Manual | Volume surrounding a large dendritic spine, an oblique and an apical dendritic segments | Harris et al. (2015) |
| Mouse: neocortex | ssTEM and FIB-SEM | FIB-SEM: 5 | – | Semi-automated | Tripartite synapses | Korogod et al. (2015) |
| Mouse: dentate gyrus | ssTEM | – | 50 | – | Tripartite synapses | Krzisch et al. (2015) |
| Mouse: frontal cortex | SBF-SEM | 4–6 | – | Manual | Tripartite synapses | Bellesi et al. (2015) |
| Rat: stratum radiatum of the hippocampal CA1 region | FIB-SEM | 6 | 5 | Semi-automated | Fully reconstructed volume: axons, dendrites, astrocytes, capillary | Cali et al. (2016) |
| Mouse: somatosensory cortex (layer I) | – | – | – | Manual | Astrocytes, glycogen granules and mitochondria | Mohammed et al. (2018) |
| Rat: various brain regions | ssTEM | – | 50–55 | Manual | Astrocytic processes containing organelles | Varela-Echevarria et al. (2017) |
| Mouse: stratum radiatum of the hippocampal CA1 region | Axial STEM tomography | 1.4 | 1.5° tilt increment | Manual | Tripartite synapses | Sweeney et al. (2017) |
| Mouse: frontal cortex (layers II–III) | SBF-SEM | 4–6 | – | Manual | Tripartite synapses | Bellesi et al. (2017) |
| Rat: stratum radiatum of the hippocampal CA1 region | ssTEM | – | 60–70 | Manual | Dendritic shafts, astrocytic processes | Gavrilov et al. (2018) |
| Mouse: somatosensory cortex (layer 2–3) | SBF-SEM and FIB-SEM | 3.854, 4.891 (FIB-SEM) | 50 (SBF-SEM), 20 (FIB-SEM) | Manual | Tripartite synapses | Lanjakornsiripan et al. (2018) |
| Mouse: somatosensory cortex (layer II-III, autism spectrum disorder model) | FIB-SEM | 5–5.58 | 40 | Manual | Tripartite synapses | Sato & Okabe (2019) |
| Human: temporal lobe neocortex (layer V) | ssTEM | – | 45–55 | Manual | Tripartite synapses | Yakoubi et al. (2019) |
| Rat: somatosensory cortex | SBF-SEM | 20 | 50 | Hybrid: manual and semi-automated | Neurons, astrocytes, microglia, pericytes, endothelial cells, and a few non-identifiable cells | Cali et al. (2019) |

(Continued)

**Table 2. (Continued)**

| Model/region | EM technique | Resolution | Section thickness (nm) | Segmentation | Labelled (sub)cellular structures | Paper |
|---|---|---|---|---|---|---|
| Mouse: hippocampal CA1 region | SBF-SEM | 6 | 50 | Hybrid: manual and semi-automated | Spine heads and post synaptic densities, glycogen granules | Vezzoli et al. (2020) |
| Rat: somatosensory cortex (layer IV) | FIB-SEM | 7 | 20 | Automated | Tripartite synapses | Kikuchi et al. (2020) |
| Human: temporal lobe of the cerebral cortex | ATUM with multibeam SEM | 8–11.1 | 3 | Automated | 50,000 cells with hundreds of millions of neurites | Shapson-Coe et al. (2021) |
| Mouse: sensorimotor cortex | ssTEM | – | 53 | Manual | Tripartite synapses and astrocytic processes | Fomitcheva et al. (2023) |
| Mouse: hippocampal CA1 region | SBF-SEM | 7.7 | 75 | Hybrid: manual and semi-automated | Astrocytes | Aten et al. (2022) |
| Mouse: visual cortex | ssTEM | 3.58 | 40 | Semi-automated | Pyramidal and non-pyramidal neurons, astrocytes, microglia, oligodendrocytes and precursors, pericytes, vasculature | Turner et al. (2022) |
| Mouse: hippocampal CA1 region | SBF-SEM | 5–7 | 75 | – | Spines, astrocytic processes | Xu et al. (2023) |
| Mouse: retina | SBF-SEM | 6 | 100–120 | Hybrid: manual and semi-automated | Capillary vasculature, astrocytes, endothelium, pericytes and neurons | Albargothy et al. (2023) |
| Mouse: visual cortex | ssTEM | 4 | 40 | - | Precapillary sphincter, astrocytes | Grubb(2023) |
| Mouse: medial nucleus of the trapezoid body | SBF-SEM | 4–10 | – | Manual | Axon and astrocytic processes with vascular endothelial cells and pericytes, oligodendrocyte | Heller et al. (2023) |
| Mouse: somatosensory cortex | FIB-SEM | 4.13 | 8 | Manual | Astrocytes and organelles | Salmon et al. (2023) |

size and team resources. Most automated methods use machine learning, often requiring extensive training data, although pre-trained models can sometimes be applied (Aswath et al., 2023; Borrett & Hughes, 2016; Peddie et al., 2022). This scoping review includes 39 papers focusing on the analysis of astrocytes ultrastructure at the vEM level (excluding the six oldest papers that were graphically reconstructed). Thirty-seven out of those detailed the image processing methods to various extents. Figure 3 illustrates the frequency of use for each software, categorized by segmentation, visualization, analysis, integrated software and custom solutions. The majority of papers used manual segmentation; the most frequently used were Reconstruct (Boston University, Boston, CA, USA) (Fiala, 2005) ($n = 11$) and TrakEM2 (Fiji plug-in) (Cardona et al., 2012) ($n = 10$), both of which can be embedded in a hybrid, semi-automated segmentation pipeline for proofreading and IMOD (Kremer, 1996) ($n = 3$), originally developed for electron tomography. Ilastik (Berg et al., 2019) ($n = 6$) includes semi-automated and automated modules, powered by machine learning and more recently a deep learning module, although it was found to be very efficient on dense segmentation of FIB-SEM stacks. For visualization, we found that Blender (Stichting Blender Foundation, Amsterdam, The Netherlands) ($n = 10$), Matlab (MathWorks, Natick, MA, USA) ($n = 2$) and Arivis Vision 4D (Zeiss, Wetzlar, Germany) ($n = 1$) were the most popular ones; although, for analysis, they were Neuromorph (Blender plug-in) (Jorstad et al., 2015) ($n = 5$), ImageJ (NIH, Bethesda, MD, USA) (Schindelin et al., 2012) (total of five; among them three used Fiji, a distribution of ImageJ with plugins for scientific image analysis) and Matlab

(MathWorks) ($n = 3$). Furthermore, Amira (Thermo Fisher Scientific, Waltham, MA, USA), an integrated software for segmentation, visualization and analysis, was used in two papers. Also, several custom-made solutions were developed by researchers, particularly during the late 20th century and the early 2000s, and most of the works surveyed use more than one software package to collect and process their data. Overall, looking at these numbers, we can conclude that free platforms such as Reconstruct, Fiji (TrakEM2) and Blender emerge, at least as a first choice, as a result of their flexibility, the integration with custom codes (such as Python) and their wide distribution among specialists.

Following 3-D reconstructions, astrocytes pose visual and analytical challenges, linked to their complex morphology. Indeed, the size of perisynaptic processes is too small, whereas the number is too large to be imaged under conventional microscopy, and their full 3-D arborization is challenging to visualize and interpret. To overcome this limit, an immersive virtual reality installation known as CAVE (Cave Automatic Virtual Environment) (Calì et al., 2016) was used in an attempt to ease the visualization of these complex structures. CAVE allowed navigation into brain neuropil and parenchyma to study the spatial arrangement of astrocytic glycogen granules, compared to synapses. Preferential distribution of glycogen was inferred to be preferentially facing boutons, and this information was used to feed simulations of the glycogen shunt compared to the astrocyte-neuron lactate shuttle (ANLS) (Coggan, Keller et al., 2018) using a point model. Nevertheless, for simulations to take into account morphologies, it is important to move to geometrically accurate

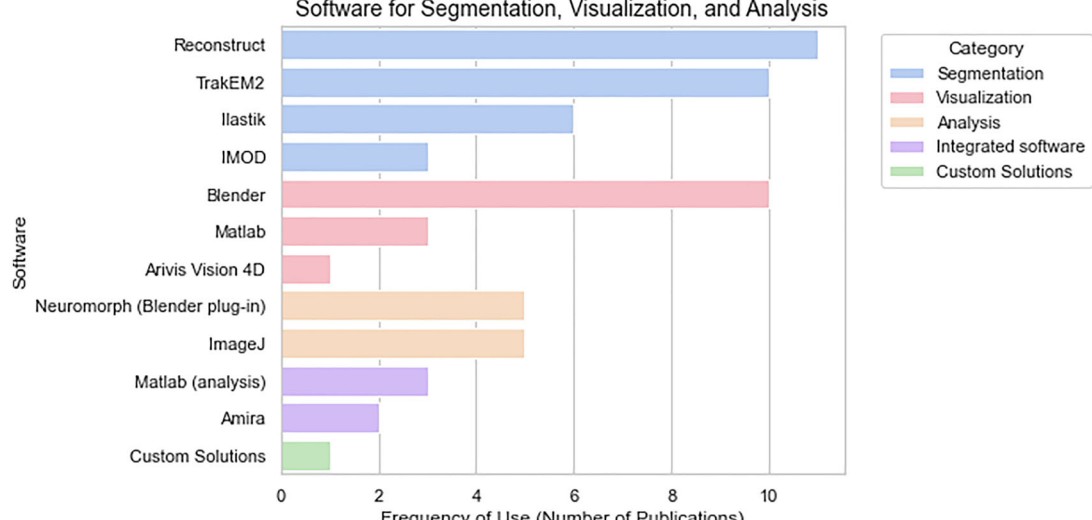

**Figure 3. Frequency of use for each software**
Frequency of use for each software extracted from the papers that reported this information and categorized by segmentation, visualization, analysis, integrated software and custom solutions.

cellular models, requiring abstract representations of the same through skeletonization of the cellular morphology. Skeletonization process of neurons is rather straightforward, but the non-Euclidean morphology of astrocytes makes it difficult to abstract their shape because of the presence of sheet-like processes and their high fractality.

This is important as skeletonization and abstraction is a step required to produce digital representation of cells prior simulation (Coggan, Calì et al., 2018; Zisis et al., 2021). In an attempt to respond to this need, an analytical tool called Abstractocyte was developed (Mohammed et al., 2018). This platform facilitates the visual examination of astrocytic processes and their interactions with neurites in the neuropil, considering non-Euclidean geometries as higher order nodes.

**CNS regions of interest in animal and human samples to analyse astrocyte morphology.** Studies from the second half of the last century until the 1990s used samples from rabbits, cats and toads, which are rarely used nowadays. To date, mouse and rat models represent the vast majority of imaged datasets as a result of their ease of maintenance and handling, as well as for ethical compliance to the 3Rs rule (replacement, reduction, refinement) compared to non-rodent mammals or primates. Some 83% of the papers included in this scoping review (Fig. 4*A*) were studies on murine samples. The inclusion criteria (vEM, astrocytes) did not encompass any work on invertebrates. However, invertebrate models have been largely studied with vEM; for example, the entire map of the nervous system of *Caenorhabditis elegans* (302 neurons and over 7000 synapses) was reconstructed already in 1986 (White et al., 1986).

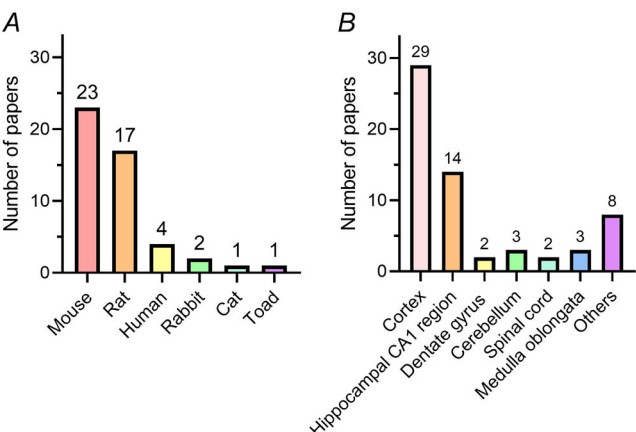

**Figure 4. Models and CNS regions of interest**
Animal models and CNS regions of interest involved in studying astrocyte morphology.

Human samples offer valuable insights. In patients undergoing invasive brain surgery (e.g. to remove tumours or drug-resistant epileptic nuclei), a small portion of normal tissue at the periphery of the lesion is usually removed and can be used for research purposes, as well as vEM imaging (Shapson-Coe et al., 2021). However, challenges in tissue preparation and authorization from ethical committees make them difficult to obtain and process. We found three papers that studied astrocyte ultrastructure using human samples; two of them analysing the temporal lobe and one the stratum radiatum of the CA1 area in the hippocampus (Shapson-Coe et al., 2021; Witcher et al., 2010; Yakoubi et al., 2019) (Fig. 5*A*). Interestingly, the majority of all the papers included in this review analysed the cortex and hippocampus (Fig. 4*B* and 5). In the latter case, only two papers have analysed dentate gyrus, whereas the remaining one focused on CA1 (Fig. 4*B*).

## Primary areas of focus encompassing the detailed study of astrocyte ultrastructure using vEM techniques

**Quantitative astroglia morphology.** Light microscopy has provided valuable information on astrocyte morphology; however, the small lamelliform processes could not be imaged using this technique. Therefore, vEM has revolutionized the study of the entire astrocyte morphology. Pioneering works from the second half of the 1900s, however, focused their investigations on individual processes, given the technical challenges of processing large volumes using ssTEM. The first paper published by Wolff (1965) investigated the morphology of the finest astrocytic processes in rat and rabbit brains; a few years later, Stensaas & Stensaas (1968) and Poritsky (1969) analysed them again, respectively, in the spinal cord of the toad and cat. A fundamental contribution in this field was given by Spacek and colleagues from the 1970s to the 2000s (Spacek, 1970; Spacek, 1985; Spacek & Harris, 1998; Spacek & Harris, 2004; Spacek & Lieberman, 1971). He focused on PAPs in the cortex and cerebellum and studied the extent of glial coverage on dendritic spines. One interesting observation was that astrocytic processes share dimensions and internal morphology of dendritic spines. During the 1990s, the first papers were published showing that astrocytic modulatory activity is mediated by calcium dynamics (Bezzi et al., 1998; Cornell-Bell et al., 1990; Parpura et al., 1994). In 1999, calcium microdomains were identified for the first time in cerebellar Bergmann glia, in a very complex ultrastructural study using ssTEM (Grosche et al., 1999). In particular, microdomains were identified as long units linked to a cabbage-like head. Despite efforts to identify visually morphological domains, the only

identifiable astrocytic processes were the cell body and the perivascular process (Calì et al., 2019). To date, the remaining arborization patterns are defined as the main branches, branchlets and leaflets, which can be defined on a qualitative and quantitative basis (Denizot et al., 2022; Gavrilov et al., 2018). Branches originate from the soma and represent the major processes, whereas branchlets are finer structures, extend from the branches, and can be secondary, tertiary or higher-order structures. The thinnest parts of the branches and branchlets, known as 'leaflets', are the regions in contact with synapses (Baldwin et al., 2023). Other interesting structures are the reflexive loop-like processes in the terminal processes, which might be related to calcium microdomains and synaptic tuning (Aten et al., 2022) (Fig. 6*A*). Moreover, these reflexive processes consistently wrap around axonal and dendritic elements, suggesting a role in providing structural support for neurites. To solve the process classification problem, Salmon et al. (2023) proposed breaking down astrocytes into essential components (defined as constrictions, expansions and cores) using a fast-marching method. They considered that the traditional classification of processes fails to recapitulate the real pattern of constrictions and expansions they identified, thus underestimating the intricate hierarchical structure of astrocytes (Salmon et al., 2023). In 2004, Hama et al. (2004) evaluated, for the first time, the surface-to-volume ratio of an individual astrocyte using electron tomography, which was estimated as $26.2 \pm 5.0$ per μm. In a more recent study, Calì et al. (2019) imaged

a volume of 750,000 μm$^3$, classifying 22 astrocytes and measuring the nuclei volume ($349.3 \pm 18.88$ μm$^3$, $N = 12$) and the surface-to-volume ratio of the four astrocytes reconstructed among the 22 identified ($4.39 \pm 0.3$ μm$^{-1}$, $N = 4$). The GFAP-like morphology was reconstructed from 3-D volume data using a thinning algorithm, enabling detailed visualization of astrocyte somas and primary branches (Fig. 5*B*). It was found that each reconstructed astrocyte had four primary branches and that the number of secondary processes correlated with cell size. Recent advancements indicate that protoplasmic astrocytes within the rodent CNS exhibit regional and layer-specific diversity (Haim & Rowitch, 2016; Khakh & Deneen, 2019). Figure 5 shows astrocyte reconstruction from layers I and V of the human temporal lobe (Shapson-Coe et al., 2021). Compared to layer VI of the rat somatosensory cortex (Fig. 5*B*) (Calì et al., 2019), human astrocytes show similar complexity but are bigger in size and their territories overlap. Interestingly, in the superficial parts of human layer I, astrocytes were smaller, higher in density and more intermingled. However, even if many papers have focused on analysing general aspects of their morphology and structural neuroglia relationships across various brain regions (Heller et al., 2023; Kikuchi et al., 2020; Rollenhagen et al., 2015; Yakoubi et al., 2019), only one has investigated the heterogeneity between layers. In the mouse somatosensory cortex, distinct morphological variations are observed among astrocytes in different layers. Specifically, research conducted by Lanjakornsiripan et al. (2018) revealed that astrocytes

*A* Human astrocytes from cortical layers I and V

*B* Rat astrocytes from cortical layer VI

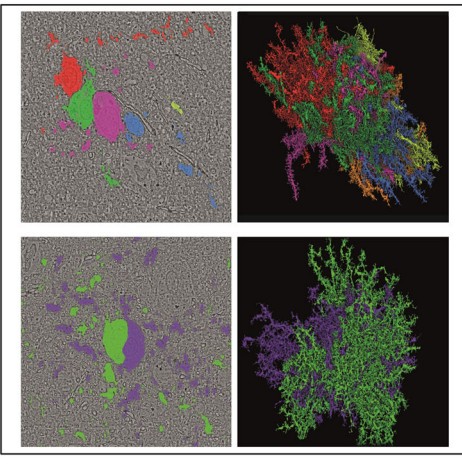

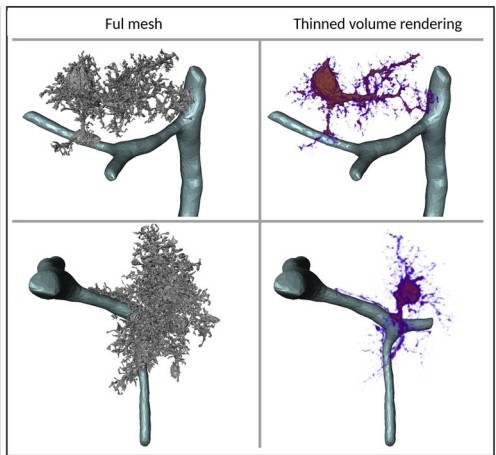

**Figure 5. Part A: Human Astrocytes from Cortical Layers I and V. Part B: Rat astrocytes from cortical layer VI**

*A*, top: six astrocytes from layer I of the human temporal lobe forming a cell aggregate with intermingling arbours, visualized through a 3-D rendering. On the bottom, two astrocytes from layer V of the human temporal lobe with closely connected cell bodies and overlapping territories. *B*, morphology of astrocytes from layer VI of the rat somatosensory cortex. Includes a full mesh rendering derived from volume segmentation and a volume rendering from a thinning procedure to emphasize primary processes. (*A*) and (*B*) are adapted from (Calì et al., 2019; Shapson-Coe et al., 2021) and reproduced under the Creative Commons Attribution license.

in layer II/III of the somatosensory cortex tend to elongate radially, whereas those in layer VI display a preference for tangential elongation. It was suggested that diversity could be a result of non-uniform extrinsic signals originating from neurons of different layers. Moreover, only a fraction of synapses is enveloped by astrocytes, with a higher proportion found in layers II/III (∼80%) compared to layer VI (∼40%) (Lanjakornsiripan et al., 2018). Cross-regional disparities have also been

documented: the rat neocortex has a relatively low percentage of excitatory synapses ensheathed by astrocytic processes (Bernardinelli et al., 2014), contrasting with the hippocampal CA1 stratum radiatum, where PAPs are present in many synapses (∼57.62%) (Ventura & Harris, 1999; Witcher et al., 2007).

**Neuropil-located processes.** Synaptic plasticity enables the consolidation of relevant information as well as the

### Different Research Focus Encompassing Astrocyte Ultrastructure

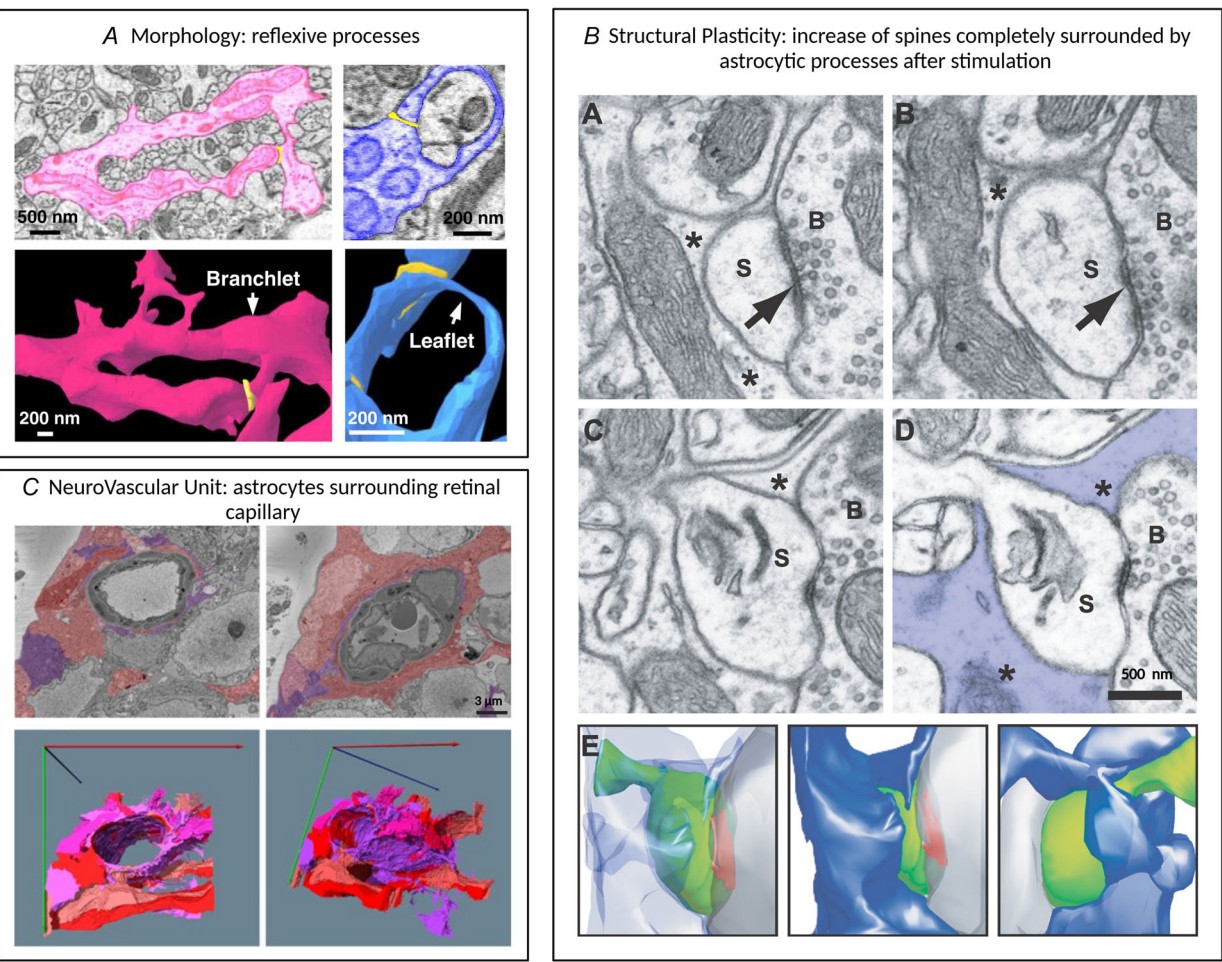

**Figure 6. Papers related to different questions regarding astrocyte ultrastructure**
Three examples of papers related to different biological questions regarding astrocyte ultrastructure. *A*, detailed astrocyte morphology is shown. Left: an electron microscopy image and a 3-D reconstruction of an astrocyte branchlet that divides into two processes, creating reflexive contacts near the endfoot process. Reflexive contacts are highlighted in bright yellow. Right: another example of an astrocytic leaflet process is depicted, looping back to the branchlet. A reflexive contact near the loop's apex is visible but does not form a closed loop structure. *B*, four electron micrographs and the 3-D reconstruction of an entire synapse with the dendritic spine (S) and the bouton (B). The perisynaptic astrocytic process that ensheath the spine is marked with an asterisk (*). Dendritic spines where the astrocytic element completely surrounds the bouton–spine interface, exhibit a significant increase in number in mice undergoing whisker stimulation. *C*, the neurovascular unit morphology is reconstructed. Segmented data and 3-D reconstructions highlight the intricate organization of macroglia surrounding the vasculature. Macroglia (red and purple) wrap around capillaries at varying depths, with astrocytes specifically identified in purple. (*A*), (*B*) and (*C*) are adapted from (Albargothy et al., 2023; Aten et al., 2022; Genoud et al., 2006) and reproduced under the Creative Commons attribution license.

removal of useless information to make room for newer connections. Astrocytes exhibit structural plasticity in response to synaptic activity, and this dynamic behaviour contributes to the remodelling of nearby synapses. For example, using a protocol of whisker stimulation, which is known to induce long-term potentiation at the level of the barrels of mice somatosensory cortex, it has been shown that synaptic potentiation significantly enhances astrocytic ensheathment of excitatory synapses. Specifically, PAPs were shown to protrude towards dendritic spines (unstimulated, 90.5%; stimulated 94.9%) and increased ensheathment around the post-synaptic density of ∼12%. This effect correlated with elevated expression of glutamate transporters (Genoud et al., 2006) (Fig. 6*B*). Similarly, Witcher et al. (2007) compared PAPs of perfusion-fixed hippocampal sections and fresh hippocampal slices that had experienced significant synapse loss followed by recuperative synaptogenesis. Their findings revealed that, in the mature hippocampus, synaptic astroglia coverage was less extended, compared to stimulated brain sections. PAPs were more commonly found in proximity to larger, hence more mature, excitatory synapses, suggesting that astroglia ensheathment facilitated or enhanced synapse stabilization and functioning. In the adult hippocampus, astrocytic processes were discovered to ensheath both afferent and efferent synapses of newly generated neurons, regardless of the age of the neurons or the size of their synapses. Analysing the gliogenesis and the arrangement of PAPs indicated that most of these processes originate from existing, mature astrocytes (Krzisch et al., 2015). Studies on regions other than the cortex, or the hippocampus, which are strictly related to learning and memory, confirm that astrocytic coverage is not a region-specific property, but rather generally correlated with synaptic plasticity. Ostroff et al. (2014) found that, through associative learning, synaptic plasticity and morphological alterations occur at synapses in the lateral amygdala of adult rats following fear conditioning or conditioned inhibition. They observed a temporary increase in synaptic density devoid of astrocytic ensheathing following threat conditioning. Conversely, synapses covered by astrocytic processes exhibited a reduced size after conditioned inhibition. These findings suggest that interaction with astrocytic processes prevents synapse expansion during memory consolidation (Ostroff et al., 2014). In 2020, Vezzoli et al. (2020), using the novel object recognition behavioural paradigm, demonstrated that long-term memory formation in mice induces an increase in spine volume and postsynaptic density surface area, as well as the number of total presynaptic vesicles, whereas it does not affect bouton volume. The changes could be prevented by interfering with astrocytic metabolic support (Suzuki et al., 2011); indeed, learning correlated with an increase

in the size and number of astrocytic glycogen granules, which are a source of lactate for neurons (Vezzoli et al., 2020). These findings suggest that glycogen granules might be considered a marker of plasticity, considering their preferential location in close proximity to synapses (Agus et al., 2018; Calì et al., 2016). Another critical aspect related to the metabolic support of astrocytes to neurons is the capacity to reaccumulate glycogen store, a process that is known to occur during sleep (Petit et al., 2021; Stone et al., 2023). Interestingly, work by Bellesi et al. reported that sleep deprivation can be associated with a closer proximity of PAPs to the synaptic cleft.

Furthermore, prolonged sleep restriction not only causes an increase in the overall coverage of astrocytes around individual synapses, but also enhances the astrocytic surface area in the neuropil; hence the overall number of synapses covered. It is not clear how synaptic coverage affects the energetic sustain of synapses and neuronal processes, but another effect of sleep deprivation was an increase in astrocytic phagocytosis. This observation led to speculations that astrocytic phagocytosis could be then a systemic reaction to the higher synaptic activity linked with prolonged wake (Bellesi et al., 2017; Bellesi et al., 2015).

**Neuro-glia vascular unit.** Astrocytes interface with the vascular system through their perivascular end-feet, comprising the only morphologically recognizable astrocytic process (Calì et al., 2019). Hence, several papers have analysed such a unique structural hallmark coupling neurons, astrocytes and vasculature, known as the neuro-glia vascular unit (Coggan, Calì et al., 2018). An impressive work from Mathiisen et al. (2010) reconstructed and analysed the perivascular arrangement from rat CA1 hippocampus. The endfeet are reported to completely fill the volume around the vessel, without gaps between adjacent astrocytes (Mathiisen et al., 2010). Only a few processes (probably microglia) extend through the perivascular glial sheath to establish direct contact with the endothelial basal lamina. By contrast, the endfoot covering pericyte is not fully formed, allowing neuropil elements to make contact with the basal lamina surrounding this cell type (Mathiisen et al., 2010). Retinal perivascular glia ensheatment is different, where detailed data on the nanoscale spatial arrangement retinal murine neuro-glia vascular unit revealed that endothelial cells and pericytes are extensively enveloped by the basement membrane, with cellular extensions protruding through gaps in the basement membrane, facilitating consistent interactions between pericytes and endothelial cells. Notably, these interactions often involve distinctive peg-and-socket structures. Additionally, the macroglia surrounding the capillary exhibit a complex, patchwork-like organization (Fig. 6*C*). In some regions,

macroglia directly interface with pericytes, creating localized points of contact (Albargothy et al., 2023). Interestingly, in a more recent report where the sample was embedded following high-pressure freezing (HPF) and water vitrification, astrocytic vascular ensheathment appeared less narrow, with more spacing between processes and the astrocytic processes standing at a distance from the vascular endothelium (Korogod et al., 2015), raising questions about fixation-induced artefacts on the ultrastructure of the glia limiting membrane (Watanabe et al., 2010).

## Discussion

In this scoping review, we have synthesized the primary findings in the field of astrocyte ultrastructure, providing a comprehensive summary of the animal models and CNS regions of interest, the vEM techniques employed, as well as the software used for the segmentation, reconstruction and analysis of astrocytes. Analysing the literature, it is evident that most of the works related to astrocytes focus on properties revolving around synaptic activity and neuronal functioning. This is an important drive, but astrocytes could potentially undergo development and migration influenced by other chemotactic signals. For example, they can originate from radial glial cells, which initially guide neurons to their final locations before becoming astrocytes. They then grow primary processes that extend radially and subsequently branch extensively until they occupy their tiling territories. Therefore, there may be structural-developmental principles that are currently unknown and, consequently, it may be necessary to investigate their properties independently to answer questions regarding their physiology (Zisis et al., 2021). Regarding mature astrocyte morphology, recent transcriptomic investigations have delineated various astrocytic profiles, suggesting structural diversity among astrocytes across brain regions (Endo et al., 2022) and within specific regions such as the hippocampus and cortex (Batiuk et al., 2020; Viana et al., 2023). Notably, a study included in this scoping review highlighted distinct morphological features of astrocytes in the somatosensory cortex across cortical layers (Lanjakornsiripan et al., 2018). Furthermore, understanding the unique features of the human brain requires a detailed exploration of its differences from other mammals. Although much attention has been given to comparative studies of neurons, astrocytes have received less interest. Oberheim et al. (2009) reported that protoplasmic astrocytes in the human neocortex are 2.6 times larger in diameter and possess 10 times more GFAP-positive primary processes than their rodent counterparts. However, vEM datasets provide opportunities to discover even more about these differences (Fig. 5*A*) (Shapson-Coe et al.,

2021). For example, the use of vEM determined that the smaller size of astrocytes in human brain aligns with findings in the layer I of mouse cortex (Lanjakornsiripan et al., 2018; Shapson-Coe et al., 2021). Nonetheless, substantial evidence regarding ultrastructural heterogeneity across numerous brain regions is still missing and this should be crucial for unravelling their roles in brain function, including interactions with neurons at synapses. Moreover, further research into astrocyte heterogeneity promises insights into how these cells regulate functions such as myelination and remyelination in the brain (Molina-Gonzalez & Miron, 2019; Varela-Echevarrìa et al., 2017). Additionally, the precise number of synapses that can be contacted by a single astrocyte remains unclear, as does the reason why synapses are more prone to interact with astrocyte leaflets compared to other astrocyte processes. Recent discoveries have revealed that individual synapses can be contacted by multiple astrocytes, suggesting that the regulation of synaptic activity may not necessarily be carried out solely by a single astrocyte (Aten et al., 2022). However, quantifying the proportion of total synapses contacted by more than one astrocyte is still an area requiring further investigation. Furthermore, Aten et al. (2022) highlighted also the presence of reflexive loop-like processes at astrocytic leaflets (Fig. 6*A*), which may play a role in synaptic tuning by contributing to calcium microdomains and providing structural support to neurites.

Implication of astrocytes in learning, memory and neuroplasticity processes has not yet been fully investigated. Works such as that of Genoud et al. (2006), which explored how sensory input, such as whisker stimulation in mice, can influence astrocytic morphology are of fundamental importance for uncovering the dynamic role of astrocytes in synaptic plasticity.

Finally, a topic of investigation that has divided the scientific community for years is whether secretory vesicles exist in astrocytes or not. Although certain studies have indicated that astrocytes release gliotransmitters via SNARE-dependent vesicular exocytosis, conflicting results challenged this hypothesis and it is proposed that astrocytes might use lysosomes or non-vesicular pathways to release them in a non-regulated manner (Calì, 2024). Gliotransmitter release is considered to fine-tune synaptic transmission at perisynaptic sites (Bergersen et al., 2011; Bezzi et al., 2004). Considering that neurotransmitter release is a highly regulated mechanism in space and time, non-specific release pathways might not represent the best way to interact functionally with synapses. Establishing the presence of secretory vesicles in astrocytes with vEM would help support the regulated exocytosis theory. However, attempts to explore the existence of these organelles at the ultrastructural level were unconvincing and any functional evidence is correlative and not direct (Calì, 2024).

**Limitations and future work.** One of the next major steps in astrocyte research lies in addressing the significant gaps in our understanding of human astrocyte biology and its differences from rodent models. Although much of the work in neuroscience has relied on the latter, emerging evidence underscores substantial ultrastructural and functional differences between human and murine astrocytes (Li et al., 2021; Vasile et al., 2017). For example, protoplasmic astrocytes in the human neocortex are notably larger and possess more GFAP-positive processes than their murine counterparts (Oberheim et al., 2009). However, studies directly comparing astrocyte properties, such as those by Oberheim et al. (2009), are still scarce. Expanding this comparative research is crucial for understanding whether these differences translate into distinct roles in neural network regulation. To tackle these questions, future advancements in vEM technology must overcome current technical limitations. In particular, HPF already provides superior sample pre-servation compared to chemical fixation, minimizing artefacts such as extracellular space shrinkage. Indeed, chemical fixation leads to a reduction in the extracellular space, estimated at around 20% *in vivo* compared to only 1–2% in EM images (Hrabetova et al., 2018; Sykova & Nicholson, 2008). This reduction most probably occurs as a result of the swelling of astrocytic processes during perfusion. In their study using HPF, Korogod et al. (2015) observed enhanced preservation of both the extracellular space and the ultrastructure of astrocytes. This suggests that chemical fixation methods may not be ideal for conducting a thorough examination of the ultrastructure of these cells (Aten et al., 2022; Calì et al., 2019; Korogod et al., 2015). However, HPF struggles with vitrification of sections thick enough to encompass entire human astrocytes (hundreds of microns and above). Additionally, for brain samples, tissue integrity degrades rapidly as a result of anoxia during the time required for dissection and preparation (Sosinsky et al., 2008), which is even longer in the case of human specimens. Therefore, a potential future direction in the field could be the use of NHP tissue for vEM astrocytic studies. NHP tissue offers a significant advantage because of its closer resemblance to human brain architecture and is more readily available than human tissue. Additionally, the demonstrated efficacy of vEM in NHP models for studying neurons suggests its significant potential for advancing our understanding of astrocytic structure and ultrastructure (Ashaber et al., 2020; Patterson et al., 2022). Exploring this approach could bridge the gap between current animal studies and human-focused research.

The acquisition process is limited by the imaged volume and time, but the actual bottleneck of vEM is data storage and analysis. On the one hand, many laboratories have made data available online for free download, promoting reuse. However, sharing and visualizing massive datasets (terabytes to petabytes) remains challenging and solutions include chunk-based data access, precomputed multiscale pyramids and cloud-based storage. Additionally, simplified representations such as skeletons can help with data compression (Akram et al., 2018; Kanari et al., 2018; Peddie et al., 2022). On the other hand, deep learning has advanced EM volume segmentation performance, but manual proofreading is still needed for accuracy because of segmentation challenges in complex or noisy data (Januszewski et al., 2018; MICrONS Consortium et al., 2021). Current methods for automatic segmentation primarily rely on decoder-encoder convolutional neural networks. Although these approaches have shown promising results and are increasingly integrated into neuroscience research pipelines, they require large amounts of annotated data, which is very time-consuming (Aswath et al., 2023; Heinrich et al., 2021; Shapson-Coe et al., 2021). Moreover, although these techniques perform well for cellular structures with easily identifiable shapes, such as mitochondria, they struggle with tracing cells with complex morphologies such as astrocytes. Emerging technologies such as the Segment Anything Model (SAM) (Kirillov et al., 2023) offer a promising solution to these challenges. SAM leverages large datasets and advanced machine learning techniques to generalize across diverse imaging modalities, reducing the need for extensive human input and addressing structural variability (Zhang et al., 2023). This helps improve the segmentation of complex astrocyte networks, making it a valuable tool in neuroscience. Fine-tuning SAM through techniques such as transfer learning and domain adaptation could further enhance its ability to generalize EM datasets feature extraction, improving accuracy in detecting subtle structures. Nevertheless, the deep learning error-rate is still too high to reliably segment EM data without human proofreading, which remains the rate-limiting step of this approach (Holst et al., 2016).

Finally, future in-depth astrocyte studies employing a combination of light and volume EM (volume correlative light-electron microscopy, vCLEM) are essential for a thorough comparison between these two investigative methods. vCLEM can be employed to target and track a structure across two imaging modalities, or it can be used to identify molecules within the volume, offering insights into their functional roles. Rare events have the potential to be pre-identified at the light microscopy stage prior to undergo EM analysis. The continuous advancement, encompassing the refinement of specialized probes, the integration of microscopy systems and the enhancement of large-scale and vEM alongside super-resolution fluorescence microscopy, is currently facilitating the widespread adoption of vCLEM in the field of biology (Boer et al., 2015).

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

## Additional information

### Competing interests

The authors declare that they have no competing interests.

### Authors contributions

V.C, M.V.C, F.B, F.D.C, P.M, A.V, M.A and C.C were responsible for the conception or design of the work. V.C, M.V.C, F.B, F.D.C, P.M, A.V, M.A and C.C were responsible for the drafting the work or revising it critically for important intellectual content. All authors listed above have read and approved the final version of the manuscript submitted for publication. All persons designated as authors qualify for authorship, and all those who qualify for authorship are listed, and agree to be accountable for all aspects of the work.

### Funding

This work was funded by the Qatar National Library (QNL) to Marco Agus.

### Acknowledgements

We thank Joao Filipe Oliveira for his criticisms and suggestions on the manuscript.

Open access publishing facilitated by Universita degli Studi di Torino, as part of the Wiley - CRUI-CARE agreement.

### Keywords

FIB-SEM, leaflets, perisynaptic processes, SBF-SEM, ssTEM, astrocyte, ultrastucture, volume electron microscopy

## Supporting information

Additional supporting information can be found online in the Supporting Information section at the end of the HTML view of the article. Supporting information files available:

**Peer Review History**

