## [Peer Review History · The Journal of Physiology]

Ultrastructure of astrocytes using volume electron microscopy: a scoping review

Vanessa Chiappini, Maria Fernanda Veloz Castillo, Francesco Biancardi, Ferdinando Di Cunto, Pierre Magistretti, Alessandro Vercelli, Marco Agus, and Corrado Cali

DOI: 10.1113/JP287455

Corresponding author(s): Corrado Cali (corrado.cali@unito.it)

The following individual(s) involved in review of this submission have agreed to reveal their identity: Paola Bezzi (Referee #1)

Review Timeline:

Submission Date:	29-Nov-2024
Editorial Decision:	10-Jan-2025
Revision Received:	31-Jan-2025
Accepted:	12-Feb-2025

Senior Editor: Nathan Schoppa

Reviewing Editor: Valentina Mosienko

Transaction Report:

Dear Associate Professor Cali,

Re: JP-TR-2024-287455 "Ultrastructure of astrocytes using volume electron microscopy: a scoping review" by Vanessa Chiappini, Maria Fernanda Veloz Castillo, Francesco Biancardi, Ferdinando Di Cunto, Pierre Magistretti, Alessandro Vercelli, Marco Agus, and Corrado Cali

Thank you for submitting your manuscript to The Journal of Physiology. It has been assessed by a Reviewing Editor and by 2 expert referees and we are pleased to tell you that it is acceptable for publication following satisfactory revision.

ABSTRACT FIGURES: Authors may use The Journal's premium BioRender account to create/redraw their Abstract Figures (and any other suitable schematic figure). Information on how to access this account is here: <https://physoc.onlinelibrary.wiley.com/journal/14697793/biorender-access>.

REVISION CHECKLIST: Upload a full Response to Referees file. To create your 'Response to Referees' copy all the reports, including any comments from the Senior and Reviewing Editors, into a Microsoft Word, or similar, file and respond to each point, using font or background colour to distinguish comments and responses and upload as the required file type.

We look forward to receiving your revised submission.

Yours sincerely,

Nathan Schoppa

EDITOR COMMENTS

Reviewing Editor:

An excellent review highlighting advances in vEM to study astrocyte morphology and structure - please, make sure to address minor comments raised by both reviewers before the manuscript can be accepted for publication.

Senior Editor:

We are pleased to let you know that your review article has been favorably reviewed and we are offering acceptance of the manuscript pending your addressing the minor points that were raised by the reviewers.

Publication will also depend on some improvements in the writing. Just in the first two sentences of the abstract, there are grammatical errors: "The morphological features of astrocytes is crucial for brain homeostasis, synaptic activity, and 17 structural support, yet remains poorly quantified. Due to nanometer-sized cross-section of neuropil astrocytic processes, electron microscopy (EM) is the only technique to date capable of revealing their finest morphologies."

"Is" should be "Are" in first sentence. "The" is needed before nanometer in second sentence.

It is important that you thoroughly check the writing throughout the manuscript and correct such errors before this work can be published.

REFEREE COMMENTS

Referee #1:

The review from Chiappini and colleagues focuses on the ultrastructure of astrocytes using volume electron microscopy (vEM), offering a detailed exploration of the various techniques and methodologies for studying astrocytes at a cellular and structural level.

The strength of this review lies in its comprehensive summary of the state of the art in the field, highlighting the technological advancements in vEM, such as serial block-face electron microscopy (SBF-SEM) and focused ion beam scanning electron microscopy (FIB-SEM). These methods allow researchers to capture large volumes of tissue for detailed three-dimensional reconstructions, which has significantly enhanced the understanding of astrocyte morphology and their role in brain function. Additionally, the review is robust in categorizing the literature by animal models, brain regions, and vEM techniques, making it an invaluable resource for those interested in astrocyte research.

Minors

The review mentions advances in segmentation software for processing vEM images.

1. Can you provide more detail on the current limitations of these tools? How do you address the challenges of segmenting astrocyte structures in complex tissue volumes, and are there any emerging methods or technologies that show promise for overcoming these limitations?
2. What do you see as the next major step in vEM for astrocyte research? Are there specific biological questions that still require resolution, and how do you foresee future advancements in vEM technology helping to answer them?

Referee #2:

This review discusses studies employing volume electron microscopy (vEM) methods to reconstruct astrocytes. It includes 39 papers published from 1998 to 2023, selected through a thorough PubMed search. The review is interesting, timely, and easy to read. It is well-organized, beginning with a description of various vEM methodologies, which is especially useful for readers unfamiliar with these techniques. It then summarizes the main findings of the reviewed papers, focusing on quantitative analysis, morphology, and synaptic plasticity. I found this review to be very informative and enjoyable to read. I only have some minor suggestions for the authors' consideration.

1. The review notes that the vast majority of the studies reviewed were conducted in murine species. Only a few studies used human tissue, as shown in Figure 4. The authors comment on the practical and ethical challenges of acquiring human tissue for vEM studies. Interestingly, there are no studies of this type conducted in non-human primates (NHPs). I suggest that the authors include, in the discussion of possible future directions for the field, the potential use of NHP tissue for vEM astrocytic studies. Such tissue is more readily available than human tissue, and several studies already use vEM in NHP tissue to study neurons.

2. Although abbreviations are defined in the legend to the graphical abstract, I strongly encourage defining abbreviations when they first appear in the main text.

3. Figures 5 and 6 showcase beautiful examples of studies using vEM to examine astrocytes. I suggest describing these results more extensively in the text and elaborating on their implications in the overall conclusions of the review.

4. Line 201: Define ANLS.

5. Minor wording issues noted:

o Lines 89-90: The phrase "whose information might be found..." is unclear. Additionally, the word "tacked" may be a typo; could this sentence be reworded for clarity?

o Line 160: The phrase "EM moved shifted the technical..." should be revised to remove either "moved" or "shifted."

o Line 193: The sentence "the size and number of perisynaptic processes is too small" could be rephrased as "the size of perisynaptic processes is too small, while the number is too large."

REQUIRED ITEMS

- Your MS must include a complete "Additional information section" with the following 4 headings and content:

Competing Interests: A statement regarding competing interests. If there are no competing interests, a statement to this effect must be included. All authors should disclose any conflict of interest in accordance with journal policy.

Author contributions: Each author should take responsibility for a particular section of the study and have contributed to writing the paper. Acquisition of funding, administrative support or the collection of data alone does not justify authorship; these contributions to the study should be listed in the Acknowledgements. Additional information such as 'X and Y have contributed equally to this work' may be added as a footnote on the title page.

It must be stated that all authors approved the final version of the manuscript and that all persons designated as authors qualify for authorship, and all those who qualify for authorship are listed.

Funding: Authors must indicate all sources of funding, including grant numbers. If authors have not received funding, this must be stated.

It is the responsibility of authors funded by RCUK to adhere to their policy regarding funding sources and underlying research material. The policy requires funding information to be included within the acknowledgement section of a paper. Guidance on how to acknowledge funding information is provided by the Research Information Network. The policy also requires all research papers, if applicable, to include a statement on how any underlying research materials, such as data, samples or models, can be accessed. However, the policy does not require that the data must be made open. If there are considered to be good or compelling reasons to protect access to the data, for example commercial confidentiality or legitimate sensitivities around data derived from potentially identifiable human participants, these should be included in the statement.

Acknowledgements: Acknowledgements should be the minimum consistent with courtesy. The wording of acknowledgements of scientific assistance or advice must have been seen and approved by the persons concerned. This section should not include details of funding.

- Author profile(s) must be uploaded via the submission form. Authors should submit a short biography (no more than 100 words for one author or 150 words in total for two authors) and a portrait photograph of the two leading authors on the paper. These should be uploaded and clearly labelled together in a Word document with the revised version of the manuscript. Any standard image format for the photograph is acceptable, but the resolution should be at least 300 DPI and preferably more. A group photograph of all authors is also acceptable, providing the biography for the whole group does not exceed 150 words.

- It is the authors' responsibility to obtain any necessary permissions to reproduce previously published material and to list these within the main article file. For information, please see: https://jp.msubmit.net/cgi-bin/main.plex?form_type=display_requirements#permissions.

- Your MS must include a complete "Additional information section" with the following 4 headings and content:

Competing Interests: A statement regarding competing interests. If there are no competing interests, a statement to this effect must be included. All authors should disclose any conflict of interest in accordance with journal policy.

Author contributions: Each author should take responsibility for a particular section of the study and have contributed to writing the paper. Acquisition of funding, administrative support or the collection of data alone does not justify authorship; these contributions to the study should be listed in the Acknowledgements. Additional information such as 'X and Y have contributed equally to this work' may be added as a footnote on the title page.

It must be stated that all authors approved the final version of the manuscript and that all persons designated as authors qualify for authorship, and all those who qualify for authorship are listed.

Funding: Authors must indicate all sources of funding, including grant numbers. If authors have not received funding, this must be stated.

It is the responsibility of authors funded by RCUK to adhere to their policy regarding funding sources and underlying research material. The policy requires funding information to be included within the acknowledgement section of a paper. Guidance on how to acknowledge funding information is provided by the Research Information Network. The policy also requires all research papers, if applicable, to include a statement on how any underlying research materials, such as data, samples or models, can be accessed. However, the policy does not require that the data must be made open. If there are considered to be good or compelling reasons to protect access to the data, for example commercial confidentiality or legitimate sensitivities around data derived from potentially identifiable human participants, these should be included in the statement.

Acknowledgements: Acknowledgements should be the minimum consistent with courtesy. The wording of acknowledgements of scientific assistance or advice must have been seen and approved by the persons concerned. This section should not include details of funding.

- Author profile(s) must be uploaded via the submission form. Authors should submit a short biography (no more than 100 words for one author or 150 words in total for two authors) and a portrait photograph of the two leading authors on the paper. These should be uploaded and clearly labelled together in a Word document with the revised version of the manuscript. Any standard image format for the photograph is acceptable, but the resolution should be at least 300 DPI and preferably more. A group photograph of all authors is also acceptable, providing the biography for the whole group does not exceed 150 words.

- It is the authors' responsibility to obtain any necessary permissions to reproduce previously published material and to list these within the main article file. For information, please see: https://jp.msubmit.net/cgi-bin/main.plex?form_type=display_requirements#permissions.

END OF COMMENTS

Dear prof. Schoppa,

We are very grateful to the reviewer for going through our manuscript, and for their valuable comments. We will report them point-by-point, hoping that they will be satisfied and that we successfully improved the quality of the manuscript.

Looking forward to hear your final decision regarding our work.

With my best regards,

Prof. Corrado Cali
On Behalf of all authors.

Senior Editor:

We are pleased to let you know that your review article has been favorably reviewed and we are offering acceptance of the manuscript pending your addressing the minor points that were raised by the reviewers.

We thank the editor for the positive comment, and we are happy to address the minor points from the reviewer, that we will detail, point by point.

Publication will also depend on some improvements in the writing. Just in the first two sentences of the abstract, there are grammatical errors: "The morphological features of astrocytes is crucial for brain homeostasis, synaptic activity, and structural support, yet remains poorly quantified. Due to nanometer-sized cross-section of neuropil astrocytic processes, electron microscopy (EM) is the only technique to date capable of revealing their finest morphologies."

"Is" should be "Are" in first sentence. "The" is needed before nanometer in second sentence. It is important that you thoroughly check the writing throughout the manuscript and correct such errors before this work can be published.

We have gone through the manuscript to check for typos, grammatical errors, and improvements in writing. All modifications have been highlighted and are visible in the revised text.

Referee 1

The review from Chiappini and colleagues focuses on the ultrastructure of astrocytes using volume electron microscopy (vEM), offering a detailed exploration of the various techniques and methodologies for studying astrocytes at a cellular and structural level.

We thank the reviewer for the constructive comments.

1. Can you provide more detail on the current limitations of these tools? How do you address the challenges of segmenting astrocyte structures in complex tissue volumes, and are there any emerging methods or technologies that show promise for overcoming these limitations?

We have now added, as requested, a paragraph describing the current limitations of the reconstruction tools, and speculations on how to address these, specifically for astrocytes segmentation and reconstruction (line 458).

2. *What do you see as the next major step in vEM for astrocyte research? Are there specific biological questions that still require resolution, and how do you foresee future advancements in vEM technology helping to answer them?*

We have now added a paragraph (lines 421 to 449) addressing specifically the problem of one big biological question, specifically the preservation of extracellular matrix which seems to be due, in particular, to astrocytes shrinkage. Moreover, translation to non-murine astrocytes is another important aspect, which we have now added to “limitation and future work”, answering also to one question from the referee #2. As for the comment of resolution, most of scientific questions related to astrocytes ultrastructure still need to be fully addressed due to the combination of the size of smaller, perisynaptic processes of astrocytes, which are below light diffraction limit, hence requiring EM, and the large amount of data to be processed due to the high heterogeneity of these cells. This latter concept has already been explicated and discussed in the manuscript, in the introduction (line 65).

Referee 2

This review discusses studies employing volume electron microscopy (vEM) methods to reconstruct astrocytes. It includes 39 papers published from 1998 to 2023, selected through a thorough PubMed search. The review is interesting, timely, and easy to read. It is well-organized, beginning with a description of various vEM methodologies, which is especially useful for readers unfamiliar with these techniques. It then summarizes the main findings of the reviewed papers, focusing on quantitative analysis, morphology, and synaptic plasticity. I found this review to be very informative and enjoyable to read. I only have some minor suggestions for the authors' consideration.

1. *The review notes that the vast majority of the studies reviewed were conducted in murine species. Only a few studies used human tissue, as shown in Figure 4. The authors comment on the practical and ethical challenges of acquiring human tissue for vEM studies. Interestingly, there are no studies of this type conducted in non-human primates (NHPs). I suggest that the authors include, in the discussion of possible future directions for the field, the potential use of NHP tissue for vEM astrocytic studies. Such tissue is more readily available than human tissue, and several studies already use vEM in NHP tissue to study neurons.*

As previously discussed, answer to this comment was included in the answer to the second point from referee 1, specifically in line 443.

2. *Although abbreviations are defined in the legend to the graphical abstract, I strongly encourage defining abbreviations when they first appear in the main text.*

We have rechecked the manuscript thoroughly and defined the missing abbreviations in the main text (lines 115, 132, 135, 475).

3. *Figures 5 and 6 showcase beautiful examples of studies using vEM to examine astrocytes. I suggest describing these results more extensively in the text and elaborating on their implications in the overall conclusions of the review.*

We have now included in the text, both in the results and in the discussion sections, more details regarding figures 5 and 6 (Lines 278 to 287; Lines 361 to 367; Lines 391 to 399; Lines 409 to 416)

4. Line 201: Define ANLS.

Modified (Now in line 206)

5. Minor wording issues noted:

o Lines 89-90: The phrase "whose information might be found..." is unclear. Additionally, the word "tacked" may be a typo; could this sentence be reworded for clarity?

The sentence has been re-worded as follows (lines 85 to 89):

Furthermore, a growing awareness that the compartmentalization of astrocytes has fundamental functional implications strongly encourages a shift from point-based models to geometrical ones (Denizot et al. 2022). Many of the data necessary to populate such models can be found in the existing literature, following an approach already addressed by the Human Brain Project consortium.

o Line 160: The phrase "EM moved shifted the technical..." should be revised to remove either "moved" or "shifted."

We have removed "shifted".

o Line 193: The sentence "the size and number of perisynaptic processes is too small" could be rephrased as "the size of perisynaptic processes is too small, while the number is too large."

Thanks for the suggestion. It has been rephrased as suggested.

Required items

- Your MS must include a complete "Additional information section" with the following 4 headings and content: Competing Interests, Author contributions, Funding, Acknowledgements.

We have added the four sections following the journal guidelines (line 484 onward).

- Author profile(s) must be uploaded via the submission form. Authors should submit a short biography (no more than 100 words for one author or 150 words in total for two authors) and a portrait photograph of the two leading authors on the paper. These should be uploaded and clearly labelled together in a Word document with the revised version of the manuscript. Any standard image format for the photograph is acceptable, but the resolution should be at least 300 DPI and preferably more. A group photograph of all authors is also acceptable, providing the biography for the whole group does not exceed 150 words.

We have proceeded in adding a short authors biography.

- It is the authors' responsibility to obtain any necessary permissions to reproduce previously published material and to list these within the main article file. For information, please see: https://jp.msubmit.net/cgi-bin/main.plex?form_type=display_requirements#permissions.

Details on permissions to reproduce images can be found on image captions.

Dear Associate Professor Cali,

Re: JP-TR-2025-287455R1 "Ultrastructure of astrocytes using volume electron microscopy: a scoping review" by Vanessa Chiappini, Maria Fernanda Veloz Castillo, Francesco Biancardi, Ferdinando Di Cunto, Pierre Magistretti, Alessandro Vercelli, Marco Agus, and Corrado Cali

We are pleased to tell you that your paper has been accepted for publication in The Journal of Physiology.

Authors should note that it is too late at this point to offer corrections prior to proofing. Major corrections at proof stage, such as changes to figures, will be referred to the Editors for approval before they can be incorporated. Only minor changes, such as to style and consistency, should be made at proof stage. Changes that need to be made after proof stage will usually require a formal correction notice.

Yours sincerely,

Nathan Schoppa
Senior Editor
The Journal of Physiology

P.S. - You can help your research get the attention it deserves! Check out Wiley's free Promotion Guide for best-practice recommendations for promoting your work at www.wileyauthors.com/eoo/guide. You can learn more about Wiley Editing Services which offers professional video, design, and writing services to create shareable video abstracts, infographics, conference posters, lay summaries, and research news stories for your research at www.wileyauthors.com/eoo/promotion.

IMPORTANT NOTICE ABOUT OPEN ACCESS: To assist authors whose funding agencies mandate public access to published research findings sooner than 12 months after publication, The Journal of Physiology allows authors to pay an Open Access (OA) fee to have their papers made freely available immediately on publication.

You can check if your funder or institution has a Wiley Open Access Account here: <https://authorservices.wiley.com/author-resources/Journal-Authors/licensing-and-open-access/open-access/author-compliance-tool.html>.

EDITOR COMMENTS

Reviewing Editor:

Many thanks for the thorough manuscript revision and for addressing the reviewers' comments.

A few minor points: Line 358 - coma after citation is missing; Reference Shapson-Coe et al., 2021 bioarxiv - it has been published in Science in 2024; Reference Batiuk et al. 2020 is mentioned in the next but not included in the reference list.

Senior Editor:

We appreciate the significant revisions in your manuscript and believe that it is now acceptable for publication for our special issue Exciting Roles of Non-excitabile Cells in Brain and Heart. The reviewing editor noted a few minor points where corrections should be made at proof stage.